# Alkali-Grafting Proton Exchange Membranes Based on Co-Grafting of α-Methylstyrene and Acrylonitrile into PVDF

**DOI:** 10.3390/polym14122424

**Published:** 2022-06-15

**Authors:** Shufeng Li, Xuelin Li, Pengfei Fu, Yao Zhang

**Affiliations:** Key Laboratory of Advanced Textile Composites, College of Textile Science and Engineering, Tiangong University, Tianjin 300387, China; 2130010014@tiangong.edu.cn (X.L.); 2030010010@tiangong.edu.cn (P.F.); 2031010096@tiangong.edu.cn (Y.Z.)

**Keywords:** PVDF, alkali-grafting, α-methyl styrene, acrylonitrile, proton exchange membrane

## Abstract

A novel alkali-induced grafting polymerization was designed to synthesize a PFGPA proton exchange membrane based on the co-grafting of α-methyl styrene (AMS) and acrylonitrile (AN) into the poly(vinylidenedifluoride) (PVDF) membrane. Three kinds of alkali treatments were used: by immersing the PVDF membranes into a 1 M NaOH solution and mixing the PVDF powders with 16% or 20% Na_4_SiO_4_. Then, AMS with AN could be co-grafted into the PVDF backbones in two grafting solvents, THF or IPA/water. Finally, the grafted membranes were sulfonated to provide the PFGPA membranes. In the experiments, the Na_4_SiO_4_ treatments showed a greater grafting degree than the NaOH treatment. The grafting degree increased with the increasing amount of Na_4_SiO_4_. The grafting solvent also influenced the grafting degree. A 40–50 percent grafting degree was obtained in either the THF or IPA/water solvent after the Na_4_SiO_4_ treatment and the THF resulted in a greater grafting degree. FTIR and XPS testified that the PFGPA membranes had been prepared and a partial hydrolysis of the cyano group from AN occurred. The PFGPA membranes with the grafting degree of about 40–50 percent showed a better dimensional stability in methanol, greater water uptake capabilities, and lower ion exchange capacities and conductivities than the Nafion 117 membranes. The PFGPA membrane with the 16% Na_4_SiO_4_ treatment and THF as the grafting solvent exhibited a better chemical stability. The obtained experimental results will provide a guide for the synthesis of alkali-grafted PFGPA membranes in practical use.

## 1. Introduction

Recently, proton exchange membrane fuel cells (PEMFCs) have attracted more and more attention as portable, clean, and efficient electrochemical energy conversion systems due to their low operation temperature, long lifetime, and high power density [1]. The proton exchange membrane (PEM) is one of the key components of the PEMFCs, which provide the channel for the proton transportation [2]. Nafion, a perfluorosulfonic acid polymer with high ion conductivity, thermal, and mechanical stability, is the most successfully commercialized PEM [3]. However, Nafion membranes exhibit the drawbacks of high cost, low performance under the low relative humidity condition, which limits the commercial application on a large-scale [4]. Hence, many efforts have been made to develop alternative membranes as substitutes for Nafion [5] such as the organic/inorganic hybrid membranes [6,7], sulfonated aromatic polymers [8], blended Nafion composite membranes [9], and fluorine containing polymers by grafting polymerization [10]. Sulfonated aromatic polymers exhibit excellent water uptake capability, ion exchange capacity, and conductivity. Their shortcomings are the great swelling and bad dimensional stabilities, which further deteriorate the mechanical properties of the membranes in application. Fluorine containing polymers [10] including the fluorinated (PTFE, FEP, and PFA) or partially fluorinated (PVDF and ETFE) membranes possess excellent chemical and thermal stability, but never have conductive groups. Therefore, how to introduce ion exchange groups into the fluorinated polymeric matrix is the key to successfully preparing the fluorinated PEMs.

Radiation grafting is an effective way to prepare the PEMs for fuel cells [11,12,13]. The properties of the membranes can be easily controlled by modifying the grafting polymerization parameters. Styrene (St) [14], due to its low cost and adjustable ion exchange capacity by sulfonation, is an appropriate material to be radiation-grafted into the fluorinated membranes. In addition, acrylonitrile (AN), methylacrylonitrile (MAN) [15], and dimethylaminoethyl methacrylate (DMAEMA) [16] are also applied. It is noted that radiation grafting is strict to the polymerizing instrumentation due to the use of high energy γ- and X-ray, which is relatively toxic and expensive.

Recently, Qiu [17] proposed an alkali (NaOH) solution-induced grafting polymerization to synthesize the PVDF-*g*-PSSA membrane, which displayed a higher conductivity of 0.1119 S/cm at 25 °C and lower methanol permeability than Nafion. Guo [18] investigated the tetrasodium orthosilicate (Na_4_SiO_4_)-induced grafting polymerization of PVDF with a polystyrene sulfonated acid membrane. The experimental results indicated that polystyrene was easily grafted into the PVDF by an alkali solution. With the increasing content of NaOH or Na_4_SiO_4_, the proton conductivity of the PVDF-*g*-PSSA membranes were gradually increased.

Styrene, or its sulfonated derivatives, is the most commonly utilized radiation grafting monomer into fluorinated polymers. However, the main shortcoming of polystyrene derivatives is that the benzylic C–H bond is easily broken to form benzyl radicals, resulting in a subsequent chain scission by β-fragmentation [19,20]. Using fluorinated monomers (e.g., α, α, β-trifluorostyrene (TFS)) can increase the chemical stability. Nevertheless, TFS possesses low grafting kinetics [21] and sulfonation degree [22], leading to a too long reaction time and unacceptable ion exchange capacity. Using the pre-sulfonated monomers (e.g., sulfonyl fluoride containing TFS derivatives [21]) can improve the grafting kinetics, but they are too expensive to be utilized on a large-scale. Ring-substituted styrenic monomers such as *p*-methylstyrene or *p*-tert-butylstyrene have also been studied as alternative monomers in order to increase the chemical stability [23]. Theoretically, α-methyl styrene (AMS), a α-substituted styrenic derivative that is commercially available, is the most chemically stable, and the polymerized AMS showed superior stability to polystyrene under oxidized conditions [24]. However, the homopolymerization of AMS is difficult due to the low steric hindrance of the α-methyl group and low ceiling temperature [25]. Grafting of AMS with other monomers such as AN or methacrylonitrile (MAN) [26,27,28] could enhance the polymerization conversion. However, few studies have been found on the alkali grafted PVDF with AMS and its co-monomers.

In this paper, alkali-induced grafting proton exchange membranes with *co*-grafting PVDF with α-methylstyrene and acrylonitrile were reported. The effects of the types and contents of the alkalis and solvents on the grafting polymerization were investigated and the obtained PVDF-based proton exchange membranes (PFGPA) were characterized in terms of the morphology, liquid uptake capability, swelling, ion exchange capacity (IEC), conductivity, mechanical property and so on. The PFGPA membranes possess excellent dimensional stability and low swelling in methanol, indicating a potential application prospect. Due to the advantages of low cost, ease of preparation, and relative environmental-friendliness, the alkali grafting polymerization is attractive and exhibits a new approach to synthesizing PFGPA membranes.

## 2. Materials and Methods

### 2.1. Materials

Kynar^®^ polyvinylidene fluoride (PVDF) resins for batteries (Mw = 400,000~600,000) were purchased from Arkema, France. Alpha-methylstyrene (AMS) was purchased from Futian Chemical Com., Zhaoqing, China. Acrylonitrile (AN), N-methyl pyrrolidone (NMP), dibenzoyl peroxide (BPO), tetrasodium orthosilicate (Na_4_SiO_4_), 1,2-dichloroethane, tetrahydrofuran (THF), isopropanol (IPA), hydrogen peroxide, and methanol were purchased from Guangfu Fine Chemical Com., Tianjin, China. All reagents were used as received except that AN and AMS were distilled prior to use.

### 2.2. Preparation of the PFGPA Membranes

Preparation of the PFGPA membranes is shown in Figure 1. Two types of alkalis, NaOH and Na_4_SiO_4_, were used to treat the PVDF membranes.

NaOH treatment process: Certain amounts of PVDF powders and NMP solution were mixed, stirred for 2 h, and poured onto the glass plate. The obtained PVDF membranes were dried under vacuum at 60 °C for 10 h. Then, they were immersed into a 50 mL 1.0 M NaOH ethanol solution with 150 mg tetrabutylammonium bromide (TBAB) being added at 60 °C for 20 min. The treated PVDF membranes were washed by deionized water to neutral.

Na_4_SiO_4_ treatment process: Certain amounts of PVDF powders were dissolved in NMP solution containing 3 wt.% deionized water, then the Na_4_SiO_4_ powders were added. The mixture was stirred for 2 h, poured onto the glass plate and dried under vacuum to afford the Na_4_SiO_4_-treated PVDF membranes, expressed as “PVDF_Na_4_SiO_4_ percent”.

Before grafting, the alkali-treated PVDF membranes were swelled in 1,2-dichloromethane for 2 h. These were respectively immersed into a solvent, IPA/water (I/W) or THF, for grafting polymerization with AN and AMS as the *co*-monomers and BPO as the initiator. The grafting polymerization was performed at 50 °C for 60 h. The obtained PVDF-*g*-P(AMS-AN) were washed with acetone and 1,2-dichloromethane three times, 2 h each, to remove the residue reactants on the surface and dried under vacuum at 60 °C for 12 h. The grafting degree (*GD*) of the PVDF-*g*-P(AMS-AN) was calculated according to Equation (1).
(1)$$GD=\frac{m_{1}-m_{0}}{m_{0}}\times 100\%$$
where GD is the grafting degree; *m*_0_ and *m*_1_ are the masses of the PVDF membranes before and after the grafting polymerization, respectively.

Afterward, the grafted PVDF membranes were swelled in 1,2-dichloromethane for 2 h. Then, they were sulfonated in a mixture of chlorosulfonic acid/1,2-dichloromethane (5:100, *v*:*v*) at room temperature for 2 h [26] to obtain the PFGPA membranes. The PFGPA membranes were washed with deionized water to neutral and dried under vacuum. The obtained PFGPA membranes were expressed as “PFGPA_X_Y”, where X represents the NaOH or percent of Na_4_SiO_4_ and Y represents the grafting solvent, respectively.

### 2.3. SEM

Morphologies of the samples were observed on a S-4800 field emission scanning electron microscope from Hitachi, Japan. Prior to the measurement, the samples were dried in the vacuum oven at 60 °C for 24 h and sprayed with Au powders on the surfaces.

### 2.4. IR Analysis

FTIR was performed on a Nicolet iS50 spectrometer from ThermoFisher Scientific Co., Ltd, Shanghai, China. The spectra were measured in transmittance mode in a wavenumber range of 4000–600 cm^−1^ with a resolution of 8 cm^−1^.

### 2.5. XPS

The X-ray photoelectron spectroscopy (XPS) was performed with a D8 DISCOVER instrument from BRUKER, Germany, with monochromatic Al Kα radiation (1600 W, 40 KV, 40 mA) and 2*θ* range of 5–45°. The XPS data were analyzed and curve fitted by the XPS peak software.

### 2.6. Liquid Uptake Capability and Swelling Ability

The membranes were cut into the dimensions of about 2 × 2 cm^2^ and dried at 120 °C for 2 h to completely remove the absorbed water to afford the dry weight, length, and width. The liquid uptake capabilities were measured in three types of liquids: deionized water, 1 mol/L methanol aqueous water, and pure methanol, respectively. The membranes were immersed into the liquid at room temperature, taken out periodically, and the water on the surface was quickly removed using filter paper to provide the wet weight, length, and width.

The liquid uptake capability and area swelling were calculated according to Equations (2) and (3), respectively,
(2)$$Liquid\;uptake=\frac{W_{wet}-W_{dry}}{W_{dry}}\times 100\%$$
(3)$$Area\;swelling=\frac{A_{wet}-A_{dry}}{A_{dry}}\times 100\%$$
where $W_{dry}$ and $A_{dry}$ are the mass and surface area in the dry, respectively; $W_{wet}$ and $A_{wet}$ are the mass and surface area in the wet, respectively.

### 2.7. Chemical Stability

The PFGPA membranes were tailored into the dimensions of 2 × 2 cm^2^. Chemical stability tests were evaluated by the degradation experiment of immersing the membranes into 30 wt.% H_2_O_2_ aqueous solution at 60 °C for 6 h. Then, the membranes were intermittently taken out from the H_2_O_2_ solution and weighed after wiping off the water on the surface.

### 2.8. Ionic Exchange Capacity

Ionic exchange capacities (IECs) of the PFGPA membranes were determined by acid–base titration. Before the test, the dried PFGPA membrane was immersed into a 0.05 mol/L NaOH solution for 24 h. Afterward, the membrane was taken out and washed with deionized water several times. The washed water was combined and poured into the NaOH solution. Then, the NaOH solution was neutralized with a 0.06 mol/L HCl solution. IEC (mmol·g^−1^) was calculated according to Equation (4).
(4)$$\mathrm{IEC}=\frac{V_{\mathrm{NaOH}}\times C_{\mathrm{NaOH}}-V_{\mathrm{HCl}}\times C_{\mathrm{HCl}}}{W}$$
where $V_{\mathrm{NaOH}}$ and $V_{\mathrm{HCl}}$ are the volume of NaOH and HCl (mL), respectively; $C_{\mathrm{NaOH}}$ and $C_{\mathrm{HCl}}$ are the concentration of NaOH and HCl (mol·L^−1^), respectively; W is the weight of the membrane in the dry.

### 2.9. Proton Conductivity

The proton conductivity measurement was performed by a double probe method in SI 1287 AC impedance, UK. The membranes were placed between two Pt electrodes with the fixed distance under the AC voltage of 10 mV. Conductivities σ of the membranes were examined at room temperature according to Equation (5).
(5)$$\sigma =\frac{L}{R\times W\times T}$$
where R is the resistance (Ω) of the sample; T and W are the thickness and the width, respectively; L of 1.5 cm represents the distance between two Pt electrodes.

### 2.10. Mechanical Measurement

The mechanical properties of the modified PVDF membranes were measured by a strong electronic universal material testing machine at room temperature. The membranes were fixed with clamps with the distance of 25 mm. The clamps were moved at the speed of 20.0 mm/min until the membrane was broken.

## 3. Results and Discussion

### 3.1. Preparation of the PFGPA Membranes

PVDF possesses excellent chemical and thermal stability and it is hard to directly introduce the other structural units into the polymeric backbone. In order to facilitate the grafting polymerization, two types of alkalis, NaOH and Na_4_SiO_4_, were used to generate the active reaction points in PVDF. Alkali treatments affect the positions and amounts of the active points in PVDF. The NaOH treatment was performed by immersing the PVDF membranes into a 1 M NaOH solution at 60 °C for 20 min and the alkali-corrosion generally happened on the surfaces of the PVDF membranes. The Na_4_SiO_4_ treatment was performed by mixing the PVDF powders with Na_4_SiO_4_ in NMP, so that the active points were distributed evenly on the cross-section [29] of the PVDF membranes, maximizing the likelihood of the grafting polymerization.

Afterward, the active PVDF membranes were immersed into a mixture containing the *co*-monomers and the initiators to perform the grafting polymerization. The effects of the alkali, the solvents, and amounts of the monomers on the grafting degree were discussed. In the grafting polymerization, two types of grafting solvent, THF and IPA/water, were used, which influenced the solubility of the monomers and compositions, structures, and properties of the resulted polymers. In the IPA/water solvent, AMS and AN are hydrophobic and immiscible. Gubler [28] proposed that due to the poor polarity of AMS, using the IPA/water solvent (5:2, *v*/*v*) could increase the effective AMS concentration at the grafting front, resulting in a higher AMS incorporation into the graft copolymer. When an AMS molar fraction of 0.6 in the feed composition was required, the AMS:AN molar ratio of 1:1 in the graft component was obtained. In our experiments, an AMS molar fraction of 0.6 by Gubler’s method was used.

Table 1 lists the parameters of the alkali, monomers, and solvents in the grafting polymerization. When the amounts of AMS and AN were lower, the grafting degree by the NaOH treatment was about 2 percent in either the IPA/water or THF solvent (No. 1, 4). The grafting degree by the Na_4_SiO_4_ treatment increased with the increasing amount of Na_4_SiO_4_ in either the IPA/water (No. 2, 3) or THF (No. 5, 6). For the 20% Na_4_SiO_4_ treatment, the IPA/water solvent exhibited a greater grafting degree of 20 percent than 13.8 percent in THF. When the volume ratio of AMS/AN to the solvent increased to 4:1, the IPA/water solvent resulted in a greater grafting degree of about 30–50 percent (No. 7–9). For the NaOH treatment, the grafting degree exceeded 30 percent. The treatment of 16% Na_4_SiO_4_ reached a grafting degree of 49 percent, greater than 43.61 percent of the 20% Na_4_SiO_4_, manifesting a potential application prospect. When THF was used as the solvent, in order to improve the grafting degree, AlCl_3_, a Lewis acid, which had successfully improved the radiation-induced grafting polymerization of PVDF [30,31], was used to effectively catalyze the alkali-induced grafting polymerization in our experiments. The obtained grafting degree was slightly greater than that in IPA/water (No. 10–12). These experimental results indicate that the PVDF-*g*-P(AMS-*co*-AN) with a grafting degree of 40–50 percent had been successfully achieved in either the IPA/water or THF solvent system.

Finally, the PVDF-*g*-P(AMS-*co*-AN) membranes were sulfonated according to the literature [26] to obtain the resulted PFGPA membranes.

### 3.2. Characterization of the PFGPA Membranes

#### 3.2.1. Morphologies

The PFGPA membranes were obtained sequentially by alkali treatment, grafting polymerization, and sulfonation. Figure 1 reveals the SEM photographs of the PVDF membranes after alkali treatment. It was found that the pristine PVDF membrane possessed a smooth surface. After the PVDF membrane was treated by 1 M NaOH solution, they still remained smooth, but several tiny holes and cracks appeared. The PVDF membranes by the 16% Na_4_SiO_4_ treatment manifested a slightly uneven thickness and some shallow strips emerged. Treatment with 20% Na_4_SiO_4_ further magnified the nonuniformity of the thicknesses and several defects caused by mixing the Na_4_SiO_4_ were clearly observed. It was concluded that whenever NaOH or Na_4_SiO_4_ were utilized, the alkali treatments led to a structural damage of the PVDF membranes. This is because the alkali treatment destroyed the PVDF backbones, generating the active points to perform the grafting polymerization.

After alkali treatment, the PVDF membranes were graft polymerized, respectively, in two solvents, IPA/water or THF. Figure 2 shows the morphologies of the grafted PVDF membranes. In the IPA/water solvent, the grafted PVDF membranes with NaOH treatment showed a rough, uneven surface with a few holes. In comparison, the THF solvent resulted in more and greater holes, indicating a worse mechanical property. For the treatment by 16% Na_4_SiO_4_, the grafted PVDF membranes in the IPA/water manifested many thin and scaly stripes, along which a few tiny cracks were found. Treatment by 20% Na_4_SiO_4_ led to folded scaly stripes, rougher surfaces, and greater cracks. Distinctly, when THF was used as the solvent, the grafted PVDF membrane with the treatment of 16% Na_4_SiO_4_ showed a relatively smooth surface and a few holes were evidently formed. For the treatment by 20% Na_4_SiO_4_, the membranes possessed more and greater holes, but still remained smooth.

Our experimental results showed that the solvents distinctly influenced the morphologies of the grafted PVDF membranes. These could be explained by the differences in the solubilities of AN and AMS in IPA/water and THF. When the grafting polymerization was performed in IPA/water, AN and AM tended to concentrate on the side of the IPA and were grafted into the PVDF backbones. On the side of water, due to the insolubility, less AMS and AN were grafted. Concentration differences in AN and AMS caused by the heterogeneity of the IPA/water solvent brought about a nonuniform surface of the grafted PVDF membranes. When THF was used, the AN and AMS were dissolved well and the grafting polymerization could be performed evenly, which led to a smooth surface.

Finally, the grafted PVDF membranes were sulfonated by chlorosulfonic acid to afford the PFGPA membranes (Figure 3). The PFGPA surfaces in the grafting solvent of IPA/water with three alkali treatments of 1 M NaOH, 16% and 20% Na_4_SiO_4_, respectively, all showed scaly surfaces, along with longer and deeper cracks, and somewhere the scales fell. For THF as the grafting solvent, the surfaces of the PFGPA membranes presented as being more even and the holes became greater.

#### 3.2.2. IR Analysis

The IR spectrograms of the pristine, 16% Na_4_SiO_4_-treated and grafted PVDF, and the PFGPA membranes are presented in Figure 4. In the pristine PVDF membranes, absorption bands at 1400 cm^−1^, 1164 cm^−1^, and 879 cm^−1^ were the characteristic peaks of PVDF, representing the C–C, C–F, and –CH_2_ vibrations, respectively. After the treatment of 16% Na_4_SiO_4_, the absorption band of the C=C bond at 1564 cm^−1^ appeared, indicating the formation of the active points. The grafted PVDF membranes showed the appearance of absorption bands at 2235 cm^−1^, which was attributed to the C≡N bond of AN [26] and 700 cm^−1^ assigned to the aromatic C–H deformation of the mono-substituted benzene ring of AMS [16], whereas the absorption band of the C=C bond at 1564 cm^−1^ disappeared, showing that the AMS and AN had been successively grafted into the PVDF. After the grafted PVDF was sulfonated, the membranes manifested new absorption bands at 1035 cm^−1^ and 1006 cm^−1^, which were the characteristic peaks of the –SO_3_H group [16]. The absorption band at 700 cm^−1^ disappeared, confirming the sulfonation of AMS. It was also noted that in the PFGPA membranes, a lower absorption peak at 1660 cm^−1^ emerged, indicating a partial hydrolysis of the CN bond to the carboxyl group [32].

#### 3.2.3. XPS

The XPS analysis of the PFGPA membranes with the treatment of 16% Na_4_SiO_4_ is shown in Figure 5. The PFGPA membranes showed peaks of C 1s and F 1s at about 285 eV and 688 eV, attributed to the PVDF backbone [17]. The peaks of N 1s, O 1s, and S 2p at about 400 eV, 532 eV, and 168 eV, respectively, confirmed the grafting of AMS and AN into the PVDF membranes and sulfonation of the grafted PVDF membranes. Si 2p at 182 eV was also observed, indicating the residue of silicon oxide from Na_4_SiO_4_. From the N 1s core-level spectrum (Figure 5b), the N 1s spectrum of the PFGPA membrane could be curve-fitted with two peaks at the BEs of 399.89 eV and 401.55 eV due to the N≡C– and –NH–CO– groups, respectively, which demonstrated that the N≡C units had been partly protonated during the hydrolysis process to generate the amide groups [32]. The peak of O 1s at around 532 eV was curve-fitted with four peaks at the BEs of 531.65 eV, 532.10 eV, 532.90 eV, and 533.90 eV, corresponding to –SO_3_H, silicon oxide, –CONH_2_, and –COOH, respectively. This showed that a few N≡C groups in AMS were hydrolyzed into amides and carboxylic acid.

Therefore, it was concluded that PVDF-g-P(AMS-co-AN) had been successfully prepared and the sulfonic grafted PVDF membranes (PFGPA) were partly hydrolyzed. The hydrolysis of the cyano groups was due to the acid sulfonation condition, under which the N≡C groups were first hydrolyzed into the amides and then converted into the carboxylic acid [26]. The hydrolysis of the cyano groups was unexpected [26] and the sulfonation of the PFGPA membranes will be systematically investigated later.

### 3.3. Performances of the PFPGA Membranes

In our experiments, due to the higher grafting degree of 40–50 percent by 16% or 20% Na_4_SiO_4_ treatment, four types of the PFGPA membranes in either THF or IPA/water (I/W) were selected to measure their properties.

#### 3.3.1. Liquid Uptake Capability and Area Swelling

In this section, three liquids, deionized water, 1 M methanol aqueous solution, and pure methanol, were used to evaluate the liquid uptake capability and area swelling.

Figure 6 shows the water uptake capabilities and area swelling of various PFGPA membranes for 192 h. For the water uptake capabilities, the four PFGPA membranes and Nafion all peaked and then decreased during the 192 h. The PFGPA membranes obtained in the grafting solvent of IPA/water showed a greater water uptake capability than those in THF. It was noticeable that PFGPA_16%_I/W demonstrated the quickest and greatest water uptake. It reached 75 percent at 24 h, peaked to 90 percent at 96 h, and finally decreased to 50 percent at 192 h. PFGPA_16%_THF, PFGPA_20%_THF, and PFGPA_20%_I/W showed the maximum water uptake of 82, 40, and 54 percent, respectively, much greater than the 17 percent of Nafion.

With respect to the area swelling in deionized water, the four PFGPA membranes peaked and then remained almost unchangeable for 192 h except that PVDF_20%_THF showed the greatest area swelling of 15 percent at 96 h and then declined. Nafion displayed a maximum area swelling of 4.76 percent at 24 h. The PFGPA membranes grafted in THF showed a greater area swelling than Nafion. The PFGPA membranes grafted in the IPA/water solvent showed a similar area swelling with Nafion at 96 h. Afterward, the PFPGA_16%_I/W manifested an increasing area, swelling to 11.5 percent. However, the PFPGA_20%_I/W maintained a lower swelling of about 4 percent. These experimental results indicated that the IPA/water solvent provided a better dimensional stability in water than THF.

Figure 7 displays the liquid uptake capabilities and area swelling of various PFGPA membranes in deionized water, 1 M methanol aqueous solution, and methanol for 48 h. Nafion showed the greatest methanol uptake capability, then the 1 M methanol aqueous solution and the water uptake was the lowest. In contrast, the four PFGPA membranes all exhibited the greatest water uptake capabilities, then the 1 M methanol aqueous solution. The methanol uptake capabilities were the lowest, no more than half that of Nafion, indicating a significant methanol resistance.

For the area swelling, similar to the liquid uptake capability, Nafion revealed a remarkable area swelling in methanol of about 50 percent, almost 10 times as much as that in water and 5 times in the 1 M methanol aqueous solution. However, the four PFGPA membranes exhibited an area swelling in methanol of no more than 20 percent, much lower than that of Nafion. Especially for PFGPA_16%_I/W and PFGPA_20%_I/W, the area swellings were about 5 percent, indicating a better dimensional stability and promising application in direct methanol fuel cells.

#### 3.3.2. Chemical Stability

It is essential that the proton exchange membranes should maintain excellent chemical properties during the operation of the fuel cell. In this paper, the chemical properties of the four PFGPA membranes were evaluated by soaking in 30% H_2_O_2_ at 60 °C and the weight changes were determined. The radicals generated by H_2_O_2_ resulted in the degradation of the PVDF membranes. Figure 8 shows the weight changes of the four PFGPA membranes. In comparison, Nafion was also tested under the same experimental conditions. It was observed that Nafion showed a weight loss of less than 10 percent, indicating the best chemical stability. Except for PFGPA_16%_THF, the other three PFGPA membranes quickly descended to about 50 percent at 6 h and then leveled off. The PFGPA_16%_THF declined to about 80 percent at 18 h, then slowed down to about 75 percent, slightly lower than Nafion, but greater than 60 percent of AIEM-1 at 6% H_2_O_2_ solution by Hu [16].

On the other hand, the PFGPA_16%_THF and PFGPA_16%_I/W revealed different chemical stabilities. The grafted chains accounted for 33 percent of the weight due to their similar grafting degree of about 49 percent in Section 3.1. After degradation in H_2_O_2_, the weight of the PFGPA_16%_I/W decreased to about 40 percent, meaning that the grafted chains were almost completely degraded. However, for the PFGPA_16%_THF, the residue weight was still 75 percent. This indicated that only part of the grafted chains decomposed from the PVDF base and about 27 percent of the grafted chains were still retained, proving a better chemical stability.

#### 3.3.3. Ion Exchange Capability and Conductivity

The PFGPA membranes have sulfonic acid groups and possess the ion exchange capacities (IEC). The measured IEC values of the four PFGPA membranes are listed in Table 1. It was noted that the IPA/water solvent in grafting polymerization displayed slightly greater IEC values of above 0.7 mmol/g than THF. In particular, PFGPA_16%_I/W displayed the greatest IEC value of 0.89 mmol/g, close to 0.91 mmol/g of Nafion measured in our experiment, exhibiting a valuable application prospect. Whenever the grafting solvent was either THF or IPA/water, the PFGPA membranes with 20% of Na_4_SiO_4_ treatment showed an IEC value of about 0.75 mmol/g. The PFGPA_16%_THF showed the lowest IEC value of 0.56 mmol/g.

Figure 9 shows the impedance plots of the PFGPA membranes and the conductivities are listed in Table 1. The impedances of Nafion and the PFGPA membranes were mainly influenced by the charge transfer process. In our experiment, Nafion showed the measured conductivity of 0.0390 S/cm with the lower solution and charge transfer impedances than the four PFGPA membranes. In comparison, the four PFGPA membranes possessed a lower conductivity of about 0.0113–0.0151 S/cm. As far as the effects of the grafting solvents were concerned, the THF resulted in a lower solution impedance than that of IPA/water. PFGPA_16%_THF and PFGPA_20%_THF possessed a similar conductivity of 0.0133 S/cm, showing that the content of Na_4_SiO_4_ never greatly influenced the proton transfer capability. For the grafting solvent of IPA/water, the conductivity of the PFGPA membranes increased with the Na_4_SiO_4_ content.

#### 3.3.4. Mechanical Properties

So far, many of the PVDF-based proton exchange membranes by radiation grafting have been synthesized as substituents for Nafion [15,16,17,18], but few of their mechanical properties have been reported. In general, the radiation grafting destroyed the backbones of the PVDF and thus decreased the mechanical properties. The greater the grafting degree, the greater the decrease in the mechanical property. In this paper, the mechanical properties of the obtained PFGPA membranes were measured and the data are listed in Table 2. The effects of the alkali treatments and grafting solvents are discussed below.

The PEMs have to possess the excellent mechanical properties to withstand the tough operating conditions. To investigate the mechanical properties of the PFGPA membranes, the pristine and Na_4_SiO_4_-treated PVDF membranes and the NaOH-treated PFGPA membranes were used as a contrast. The pristine PVDF membranes showed an obvious plastic deformation with the broken stress of 34.07 MPa, a strain of 17.6%, and elastic modulus of 1771.37 MPa. After the PVDF was treated by Na_4_SiO_4_, the mechanical properties declined dramatically. Treatment by 16% Na_4_SiO_4_ showed a stress of 8.27 MPa and strain of 4.8 percent, a quarter of the pristine PVDF membranes. Treatment by 20% Na_4_SiO_4_ intensified the decrease with the stress of 4.62 MPa and the strain of 5.1 percent. After the grafting polymerization and sulfonation, compared with the PVDF_16%, the PFGPA_16%_I/W and PFGPA_16%_THF decreased in broken strain to about a half, indicating a worse mechanical property. The IPA/water brought about a lower decline in strain than the THF. For the PVDF_20%_THF and PVDF_20%_I/W, the membranes were too fragile to be measured under the experimental conditions.

Comparatively, the PFGPA_NaOH_THF and PFGPA_NaOH_I/W showed the broken stresses of 12.53 MPa and 7.14 MPa, and strains of 4.9% and 4.0%, respectively, which were close to PVDF_16% and greater than PFGPA_16%_THF and PFGPA_16%_I/W. These could be explained by the fact that PFGPA_NaOH_THF and PFGPA_NaOH_I/W possessed the grafting degree of about 35 percent, lower than 49 percent of PFGPA_16%_THF and PFGPA_16%_I/W, which resulted in a lower decrease in the mechanical property.

Our experiments demonstrated that the alkali treatment influenced the mechanical property of the PVDF membranes. The alkali treatment destroyed the PVDF backbone more than the grafting polymerization and sulfonation. The Na_4_SiO_4_ treatment led to a greater decrease in mechanical property than the NaOH treatment and the obtained PFGPA membranes became fragile [33].

## 4. Conclusions

The PFGPA membranes were prepared by alkali-induced grafting polymerization and subsequent sulfonation. The AMS could be grafted with AN into the PVDF base and IR and XPS analysis testified that AMS and AN were successfully grafted into the PVDF backbone and a partial hydrolysis of the CN group in AN occurred. A higher alkali content resulted in a higher grafting degree. THF as the grafting solvent could dissolve the AMS and AN, so that a smoother membrane and higher grafting degree were obtained than that in the IPA/water solvent. The PFGPA membranes exhibited a greater water uptake capability and better dimensional stability in methanol than Nafion, though they possessed lower IEC values and conductivities. The PFGPA_16%_THF membrane with a grafting degree of 49 percent manifested a slightly lower chemical stability than Nafion, indicating a promising application. It was also noted that the alkali treatment brought about a decrease in the mechanical property of the PVDF membranes and limits their practical application in fuel cells. The next research will focus on the enhancement of the conductivities of the PFGPA membranes and the improvement in the mechanical properties. The obtained experimental conclusions would be instructive to optimize the PFGPA membranes prepared by an alkali-grafted polymerization and promote their practical applications in PEMs.

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
