# Peer review of "Alkali-Grafting Proton Exchange Membranes Based on Co-Grafting of α-Methylstyrene and Acrylonitrile into PVDF"

_polymers, 2022, doi:10.3390/polym14122424_

Round 1

Reviewer 1 Report

The paper entitled “Alkali-grafting proton exchange membranes based on co-grafting of a-methylstyrene and acrylonitrile into PVDF” by Li S. et al. has been reviewed. The idea of PVDF grafting and further sulfonation seems to be very promising. But there are some remarks which should be taken into consideration before publication. To start with, few English bugs should be corrected (for example, line 57; line 213; line 239; lines 257-258; line 301; lines 317-319; etc.). 1.      Are you sure to give the grafting degree with such high precision? Besides, the grafting degree was determined only by mass weighting. However, it would be interesting to determine also the sulfur content (by XPS, for example) in order to correlate the grafting degree with the sulfonation degree.   2.      The wavenumber should be the same for the same adsorption bond in text and in Fig. 4 (for example, 847 cm-1 in text and 879 cm-1 in Fig. 4; 1560 and 1568 cm-1 in text and 1564 cm-1 in Fig. 4 for C=C bond and so on). 3.      The values in Table 2 are too precise. Could you explain how such a precision was obtained? 4.      What does it mean “bad fragility” (line 440)? 5.      As the synthesized membranes are designated to be used in fuel cells, it would be useful to measure the conductivity of such membranes. Also, the polarization curves should be measured and compared with the reference membrane.

Reviewer 2 Report

polymers-1749173

This study reports alkali-grafting Proton Exchange Membranes Based on Co-grafting of α-methylstyrene and Acrylonitrile into PVDF. It is an excellent work and I would recommend for publication.

Abstract: please provide a brief methodology.

Figures 6 and 7: the 3D figure is very hard to read. It is recommended to change to 2D.

Reviewer 3 Report

Dear Authors

In your study, A novel alkali-induced grafting polymerization was designed to synthesize a PFGPA proton exchange membrane based on the co-grafting of a-methyl styrene (AMS) and acrylonitrile (AN) into the poly(vinylidenedifluoride) (PVDF) membrane. AMS with AN could be successfully co-grafted into the PVDF backbones after the NaOH- or Na4SiO4- treatment. An increasing alkali amount resulted in an increasing grafting degree. The grafting solvent, THF or IPA/water, also influenced the grafting degree. A 40-50 percent grafting degree was obtained in the grafting solvent of either THF or IPA/water after the Na4SiO4 treatment. FTIR and XPS testified that the PFGPA membranes had been prepared and a partial hydrolysis of the cyano group from AN occurred. The PFGPA membranes with the grafting degree of about 40-50 percent showed the better dimensional stability in methanol and greater water uptake capabilities than Nafion 117 membranes. The PFGPA membrane with the 16% Na4SiO4 treatment and THF as the grafting solvent showed the better chemical stability. Though the PFGPA membranes possessed a lower ion exchange capacities and conductivities, due to the advantages of low cost, ease of preparation and relatively environmental friendship, the PFGPA prepared by the alkali-induced grafting provides a novel solution to the candidate for Nafion.

The work presented is interested for the readers and has some novelty. 

However, some comments and concerns are remaining. You can take into consideration during the revision of your manuscript.

Comments

The detailed names of some abbreviations must be mentioned with the first time of appearance in the text. For example; PFGPA and IPA.

Scheme 1: AIBN is indicated as initiator while the authors mentioned that they are using BPO as initiator in the text and table 1.

The authors dismissed the study of the sulfonation step conditions which is very essential to have membranes with high ionic conductivity. The authors need to study the sulfonation process affecting factors.

Moreover, the mechanical properties of the developed membranes are not in favor of application the membranes in the fuel cell. An improvement of the mechanical properties to an acceptable level is needed first before recommending the development membranes for application in the fuel cell. 

Accordingly, a major revision is needed before considering your manuscript for publication., 

Round 2

Reviewer 3 Report

Dear Authors 

Thanks for considering the comments during the 

Revision process of your manuscript.

I can recommend the manuscript current version for 

Publication.